# Genome-wide identification, structural and gene expression analysis of the nitrate transporters (*NRTs*) family in potato (Solanum tuberosum L.)

**Jingying Zhang, Zhijun Han, Yue Lu, Yanfei Zhao, Yaping Wang, Jiayue Zhang, Haoran Ma, Yu Zhu Han** *

College of horticulture, Jilin Agricultural University, Changchun City, P.R. China

* hanyuzhu@jlau.edu.cn

**Data Availability Statement:** All relevant data are within the paper and its Supporting information files.

## Abstract

Nitrogen ($N_2$) is the most important source of mineral N for plant growth, which was mainly transported by nitrate transporters (*NRTs*). However, little is known about the *NRT* gene family in potato. In this study, *StNRT* gene family members were identified in potato. In addition, we performed *StNRT* subfamily classification, gene structure and distribution analysis, and conserved domain prediction using various bioinformatics tools. Totally, 39 *StNRT* gene members were identified in potato genome, including 33, 4 and 2 member belong to *NRT1*, *NRT2*, and *NRT3*, respectively. These 39 *StNRT* genes were randomly distributed on all chromosomes. The collinearity results show that *StNRT* members in potato are closely related to *Solanum lycopersicum* and *Solanum melongena*. For the expression, different members of *StNRT* play different roles in leaves and roots. Especially under sufficient nitrogen conditions, different members have a clear distribution in different tissues. These results provide valuable information for identifying the members of the *StNRT* family in potato and could provide functional characterization of *StNRT* genes in further research.

## Introduction

Nitrogen (N) play an essential role that affects plant growth and development, which is an important component of chlorophyll, amino acids, nucleic acids, and secondary metabolites [1]. Nitrate ($NO_3^-$) is the most important source of mineral N for plants growth [2]. $NO_3^-$ can act as a signaling molecule that regulates gene expression in many processes, such as plant growth, root system architecture development [3], leaf growth and development [4,5], seed dormancy [6], and flowering time [7]. Plants can uptake $NO_3^-$ from soil and store it in vacuoles through a series of transport pathways [8–10], but mainly performed by nitrate transporters (*NRTs*) that are encoded by a multigene family [11]. According to their affinity for the substrate, NRTs are divided into two systems: the low-affinity transport system (LATS) (via nitrate transporter 1 family, *NRT1*) [12,13] and the high-affinity transport system (HATS) (via nitrate transporter 2 family, *NRT2*) [14,15]. Therefore, plants had evolved a series of *NRT* gene family

**Funding:** The author thanks Jilin Agricultural University Potato Innovation Team platform.The research was funded Jilin Potato Genetic breeding and Improved seed Breeding Innovation team 20200301025RQ.

**Competing interests:** The authors have declared that no competing interests exist.

members to make better use of $NO_3^-$. There were three *NRT* gene subfamilies: *NRT1*, *NRT2*, and *NRT3* [16]. Till now, several studies have elucidated *NRT* genes functions and evolutionary history in many plant species such as *Arabidopsis thaliana* [17,18], rice [19], poplar [20] and pineapple [21]. Our previous study found that *NRT* gene responded positively to nitrogen deficiency stress [22]. Besides that, Pieczynski et al reported that some *NRT* family members were not only involved in the nitrogen transportation, but also responded to drought [23]. Phylogenetic studies have revealed that *NRT1* families gather a large number of genes and could be divided in 8 to 10 subfamilies [13,24], which had been shown to incorporate transporters not only for $NO_3^-$, but also for peptides, amino acids, nitrite, glucosinolates, abscisic acid and gibberelins [2]. As compared to *NRT1* families, *NRT2* families analyzed in various species contain a much lower number of genes. In A. thaliana, there are seven members of the *NRT2* gene family from *NRT2.1* to *NRT2.7* [17,25]. Gene structure of the *AtNRT* family members were reviewed by Okamoto, but the functions of *NRT1* and *NRT2* transporters are largely unknown [25]. Further physiological analysis is needed to understand the precise role of individual *NRT* gene, in particular for potato, because there were no systematic reports on the *NRT* gene family members in potato.

As for potato, large amount of nitrogen is needed in the growth and development. Therefore, it can provide theoretical basis for potato breeding to understand the family members of *StNRT* and their relationship. In this study, *StNRT* gene family members were identified in potato. In addition, we performed *StNRT* subfamily classification, gene structure and distribution analysis, and conserved domain prediction using various bioinformatics tools. This study could be helpful for further functional study of *StNRT* genes and molecular breeding of potato.

## Materials and methods

### Genome-wide identification of NRT proteins and genes

A total of 60 *AtNRT* family members sequences from *Arabidopsis thaliana* were collected from TAIR (https://www.arabidopsis.org/) and some previous studies [25,26]. Also, according to Tsay's report, 81 *OsNRTs* were collected [26]. All these collected *NRT* members were used as queries to search against sequence homologs in the potato genome from the Ensemblplants (http://plants.ensembl.org/info/website/ftp/index.html). The candidate *StNRT* members were identified using BLAST method and HMMER 3.0 software (http://hmmer.janelia.org/). Then, the candidate members were further confirmed according to Uniport database (https://www.uniprot.org/) and those without *NRT* gene annotation were discarded. To identify the domains of the candidate members, online programmes NCBI conserved domain database (CDD) (https://www.ncbi.nlm.nih.gov/cdd/Structure/cdd/wrpsb.cgi) was used with expect value <0.05 and the results were displayed with TBtools (V0.67, https://github.com/CJ-Chen/TBtools) [27].

### Chromosomal localization and gene duplication of potato *StNRT* genes

All the candidate *StNRTs* were mapped on potato chromosomes and displayed by TBtools software according to the potato *StNRT* gene positions in the annotation file from ensemble plant genome database. To identify the duplicated and tandem repeated genes, ClustalW alignment comparison of all *StNRT* members was carried out with a threshold of similarity >75% and their genomic locations. The tandem duplicated genes were restricted within the range of 100 kb distance [28].

### *StNRTs* structure, conserved domain, motif, and phylogenetic analysis

*StNRTs* structure were analyzed by aligning the coding sequence (CDS) regions to the genomic DNA sequences. The gene structure and conserved domains obtained from CDD database of all the members were displayed using the TBtools software. The motifs were predicted via the Multiple Expectation Maximisation for Motif elicitation (MEME) online tool (http://meme-suite.org/tools/meme). As for molecular weight (MW) and the theoretical isoelectric point (pI) prediction, the online tool ExPASy (https://www.expasy.org/tools/) were used basing on the proteins sequence of all the *StNRT* members.

### Phylogenetic tree construction

To evaluate the evolution relationship of all the family members of *StNRTs*, phylogenetic tree was constructed via MEGA (version 7.0.26). Firstly, the ClustalX program was used to perform multiple sequence alignments of the *StNRTs* of *Arabidopsis thaliana* and potato. Then, Maximum Likelihood (ML) tree was constructed basing on the optimal model prediction results with 1000 bootstrap tests.

### Identification of gene synteny

Gene synteny analysis were performed by BLAST and the Multiple Collinearity Scan toolkit (MCScanX) [29] according to Song's report [30]. Briefly, the sequence of potato candidate gene family members were searched against itself using BLASTP with an E-value cut-off of $1 \times 10^{-10}$ and identity >75%. Then, the acquired BLASTP results were next used as the MCScanX input to assess the collinear blocks. For the gene synteny among different genomes, we selected 4 plant genomes for collinear analysis, including *Arabidopsis thaliana*, *Oryza sativa*, *Solanum lycopersicum* and *Solanum melongena*. The assembly of *Arabidopsis thaliana*, *Oryza sativa* and *Solanum lycopersicum* were obtained from Ensemblplants (http://plants.ensembl.org/info/website/ftp/index.html) and the assembly sequence of *Solanum melongena* was obtained from China National Genebank (CNGB, https://www.cngb.org/index.html). The analysis process refered to the instruction of MCScanX software.

### Transcriptome expression analysis

The Illumina RNA-seq data were downloaded from the SRA database (https://submit.ncbi.nlm.nih.gov/subs/sra/) with the submission number of SRS4186597 (the data was up loaded in our previous study [22]) to study the expression patterns of all the identified *StNRT* members in response to nitrogen deficiency. Briefly, Potato cultivar cv. *Shepody* was treated with sufficient-N- and deficient-N-fertilizer. Then, leaf and root transcriptomes were analyzed and differentially expressed genes (DEGs) in response to N deficiency were identified. We compared the expression differences of these *StNRT* members between the sufficient N fertilizer group and the deficient N group in leaf and root. The sequence data used was obtained from Solanum tuberosum *cv*. *Shepody*. The expression of *StNRT* members were showed in a heatmap via TBtools software.

### Cis-element enrichment analysis

The upstream sequences (2 kb) of the *StNRT* sequences were retrieved and then submitted to PlantCARE (http://bioinformatics.psb.ugent.be/webtools/plantcare/html/) to identify six regulatory elements, abscisic acid (ABA)-responsive elements (ABRE), ACE, CAT-box, estrogen response element (ERE), MYB and MYC. Then GSDS 2.0 (http://gsds.cbi.pku.edu.cn/index.php) was used to plot the location of these elements.

# Results

## Identification and analysis of *StNRT* genes

A total of 46 and 47 *StNRT* peptides sequence obtained in BLAST and HMMER3 analysis results, respectively, of which 46 members were the common genes. According to the annotation information of uniport database, all these 44 genes belong to *NRT* family. After removing the duplicate sequence, 39 genes were obtained. All these 39 sequences were reserved and submitted to CDD to confirm the conserved domain. The results showed that nine domains were identified and seven of them were MFS-related domains. These 39 sequences were named based on their chromosomal locations (Table 1). The lengths of the *StNRT* proteins ranged from 203 (*StNRT07*) to 653 amino acids (*StNRT33*) with mean length of 559.10. The conserved domain results showed that the *StNRT* genes in potato contained the same domains with that of *Arabidopsis thaliana* and *Oryza sativa* (S1a and S1b Fig) and most of the genes contained complete domains (Fig 1). The molecular weights of *StNRT* genes were between 22.65 kDa (*StNRT07*) and 71.9 kDa (*StNRT33*). Theoretical pI value range from 6.03 (*StNRT20*) to 9.36 (*StNRT34*).

## Phylogenetic analysis of potato *StNRT* genes

To decipher the evolutionary relationships and functional associations of *NRT* genes in potato, the multi-species phylogenetic tree was constructed based on the full-length amino acid sequences of NRTs from potato, *Arabidopsis thaliana*, and rice with the Maximum Likelihood method. In total, 60 sequences from *Arabidopsis thaliana*, 81 sequences from rice, 39 sequences from potato were assessed in the phylogenetic tree (Fig 2). The phylogenetic analysis revealed that all the *NRTs* could be divided into four groups: *NRT1*, *NRT2*, and *NRT3.1* and *NRT3.2*. There were 33 *StNRT* genes belong to *NRT1*. There were 4 *StNRT* genes belong to *NRT2*, including *StNRT04*, *StNRT17*, *StNRT32* and *StNRT34*. In addition, we identified two *StNRT3* gene: *StNRT06* (*StNRT3.2*) and *StNRT07* (*StNRT3.1*). In addition, we found that *StNRT* genes in potato prefers to cluster with the *AtNRT* genes of *Arabidopsis thaliana*, rather than *Oryza sativa*.

## Chromosome localization and duplication of the *StNRT* gene family

The location analysis showed that all these 39 *StNRT* members were randomly distributed on the 12 potato chromosomes (Fig 3). Chromosomes 02, 07 and 08 contain two *StNRT* genes, while chromosome 03 and 06 contains the most *StNRT* genes (5 in each) among all potato chromosomes. Gene duplication events have driven the expansion of potato *StNRT* genes, with 13 genes found in 6 duplicated blocks and 26 *StNRT* genes located outside of the duplicated blocks. Six pairs of genes, including *StNRT*06/−07, 08/−09/−10, 12/−13, 15/−16, 27/−28, and 37/−38 were separated by less than a 100-kb region on chromosome 03, 03, 04, 05 and 09 and 12, respectively.

## Gene structure and motifs in *StNRT* gene family

Conserved motifs were analyzed for all the 39 *StNRT* members using MEME software and 10 motifs were identified (Fig 4a). There were no motifs found on *StNRT04*, *StNRT06*, *StNRT07* and *StNRT34*. Only one motif found on *StNRT17* (Motif 2) and *StNRT32* (Motif 2). Interestingly, these five genes mentioned above contained the PLN00028 domain (the typical characteristics of *NRT* gene). To identify the motifs that contained PLN00028 domain, we further compared the gene sequences of *Arabidopsis thaliana* and potato. The results showed that these genes in *Arabidopsis thaliana* and potato had the consistent motifs (S2 Fig). For genes

**Table 1. *StNRT* genes identified in potato and their sequence characteristics.**

| StNRT ID | Protein ID | Gene ID | Chromosomal localization | | Gene length (bp) | Amino acid length (aa) | pI* | MW (kD)* | CDS length (bp)* |
|---|---|---|---|---|---|---|---|---|---|
| *StNRT*01 | PGSC0003DMP400002994 | PGSC0003DMG400001671 | 67991306 | 67996008 | 4702 | 554 | 6.39 | 66441.93 | 1665 |
| *StNRT*02 | PGSC0003DMP400047815 | PGSC0003DMG402027501 | 74630384 | 74633515 | 3131 | 593 | 9.12 | 63512.61 | 1782 |
| *StNRT*03 | PGSC0003DMP400036006 | PGSC0003DMG400020708 | 79794672 | 79799593 | 4921 | 530 | 8.8 | 67005.04 | 1593 |
| *StNRT*04 | PGSC0003DMP400012236 | PGSC0003DMG400006913 | 26506508 | 26508700 | 2192 | 512 | 9.03 | 45858.84 | 1539 |
| *StNRT*05 | PGSC0003DMP400002522 | PGSC0003DMG400001393 | 46706587 | 46710386 | 3799 | 596 | 7.99 | 57116.7 | 1791 |
| *StNRT*06 | PGSC0003DMP400031706 | PGSC0003DMG400018193 | 51862141 | 51866797 | 4656 | 584 | 6.23 | 35792.95 | 1755 |
| *StNRT*07 | PGSC0003DMP400031592 | PGSC0003DMG400018129 | 51871879 | 51873422 | 1543 | 590 | 9.29 | 22651.25 | 1773 |
| *StNRT*08 | PGSC0003DMP400043961 | PGSC0003DMG400025339 | 53135911 | 53138548 | 2637 | 552 | 8.63 | 66220.39 | 1659 |
| *StNRT*09 | PGSC0003DMP400043959 | PGSC0003DMG400025337 | 53160486 | 53162884 | 2398 | 424 | 9.02 | 65517.71 | 1275 |
| *StNRT*10 | PGSC0003DMP400043958 | PGSC0003DMG400025336 | 53169753 | 53173630 | 3877 | 589 | 9.32 | 66164.32 | 1770 |
| *StNRT*11 | PGSC0003DMP400005176 | PGSC0003DMG400002865 | 388737 | 394987 | 6250 | 582 | 8.99 | 64393.43 | 1749 |
| *StNRT*12 | PGSC0003DMP400020732 | PGSC0003DMG400011693 | 68832667 | 68834950 | 2283 | 585 | 9.09 | 65181.01 | 1758 |
| *StNRT*13 | PGSC0003DMP400020731 | PGSC0003DMG400011692 | 68837821 | 68840580 | 2759 | 590 | 9.15 | 64547.23 | 1773 |
| *StNRT*14 | PGSC0003DMP400025609 | PGSC0003DMG400014539 | 2690484 | 2695466 | 4982 | 581 | 8.86 | 65222.23 | 1746 |
| *StNRT*15 | PGSC0003DMP400030769 | PGSC0003DMG400017620 | 6018942 | 6027657 | 8715 | 585 | 8.99 | 68498.12 | 1758 |
| *StNRT*16 | PGSC0003DMP400030807 | PGSC0003DMG400017637 | 6118146 | 6121903 | 3757 | 553 | 8.4 | 67732.91 | 1662 |
| *StNRT*17 | PGSC0003DMP400029708 | PGSC0003DMG400016996 | 9714775 | 9717165 | 2390 | 580 | 8.81 | 54353.09 | 1743 |
| *StNRT*18 | PGSC0003DMP400044048 | PGSC0003DMG400025395 | 8311895 | 8316339 | 4444 | 500 | 8.35 | 68301.19 | 1503 |
| *StNRT*19 | PGSC0003DMP400064739 | PGSC0003DMG400042635 | 31438515 | 31443669 | 5154 | 608 | 8.72 | 65380.81 | 1827 |
| *StNRT*20 | PGSC0003DMP400054567 | PGSC0003DMG401031322 | 47048581 | 47054364 | 5783 | 627 | 6.03 | 62394 | 1884 |
| *StNRT*21 | PGSC0003DMP400011673 | PGSC0003DMG400006606 | 56077004 | 56080118 | 3114 | 606 | 8.55 | 60265.93 | 1821 |
| *StNRT*22 | PGSC0003DMP400052968 | PGSC0003DMG400030438 | 57021923 | 57025262 | 3339 | 203 | 8.12 | 64066.11 | 612 |
| *StNRT*23 | PGSC0003DMP400039866 | PGSC0003DMG400022993 | 5775617 | 5778627 | 3010 | 319 | 8.87 | 65275.63 | 960 |
| *StNRT*24 | PGSC0003DMP400001318 | PGSC0003DMG402000668 | 45033113 | 45036254 | 3141 | 566 | 9.16 | 65119.92 | 1701 |
| *StNRT*25 | PGSC0003DMP400022078 | PGSC0003DMG400012479 | 479008 | 483551 | 4543 | 605 | 8.85 | 65321.31 | 1818 |
| *StNRT*26 | PGSC0003DMP400008499 | PGSC0003DMG400004795 | 52567376 | 52571923 | 4547 | 584 | 9.15 | 65171.39 | 1755 |
| *StNRT*27 | PGSC0003DMP400024739 | PGSC0003DMG400014054 | 395745 | 398087 | 2342 | 597 | 8.37 | 63771.68 | 1794 |
| *StNRT*28 | PGSC0003DMP400035282 | PGSC0003DMG400020318 | 402706 | 405477 | 2771 | 591 | 8.66 | 63508.52 | 1776 |
| *StNRT*29 | PGSC0003DMP400030052 | PGSC0003DMG400017204 | 56806392 | 56815251 | 8859 | 585 | 9.31 | 68098.05 | 1758 |
| *StNRT*30 | PGSC0003DMP400041720 | PGSC0003DMG400024120 | 37692715 | 37696235 | 3520 | 595 | 6.57 | 66177.4 | 1788 |
| *StNRT*31 | PGSC0003DMP400019579 | PGSC0003DMG400011085 | 54161108 | 54166759 | 5651 | 555 | 9.15 | 66165.74 | 1668 |
| *StNRT*32 | PGSC0003DMP400064264 | PGSC0003DMG400042160 | 55059494 | 55061848 | 2354 | 611 | 9.04 | 57804.32 | 1836 |
| *StNRT*33 | PGSC0003DMP400048707 | PGSC0003DMG400028026 | 19452106 | 19456164 | 4058 | 599 | 9.06 | 71898.12 | 1800 |
| *StNRT*34 | PGSC0003DMP400002117 | PGSC0003DMG400001145 | 41898143 | 41899905 | 1762 | 575 | 9.36 | 57595.05 | 1728 |
| *StNRT*35 | PGSC0003DMP400044035 | PGSC0003DMG400025389 | 44400838 | 44404224 | 3386 | 653 | 9.03 | 61353.52 | 1962 |
| *StNRT*36 | PGSC0003DMP400000603 | PGSC0003DMG400000303 | 4387465 | 4391552 | 4087 | 574 | 8.44 | 62005.02 | 1725 |
| *StNRT*37 | PGSC0003DMP400025448 | PGSC0003DMG400014449 | 48920928 | 48925553 | 4625 | 568 | 7.03 | 64698.13 | 1707 |
| *StNRT*38 | PGSC0003DMP400025503 | PGSC0003DMG400014473 | 48944132 | 48959296 | 15164 | 534 | 8.08 | 61286.58 | 1605 |
| *StNRT*39 | PGSC0003DMP400045945 | PGSC0003DMG400026455 | 50643015 | 50647292 | 4277 | 570 | 8.97 | 66547.84 | 1713 |

*pI, Isoelectric point; MW (kD), Molecular weight; CDS length (bp), Coding DNA Sequence length.

For the column chromosomal localization, the number in the left-hand list is the starting position and the right-hand list is the end position.

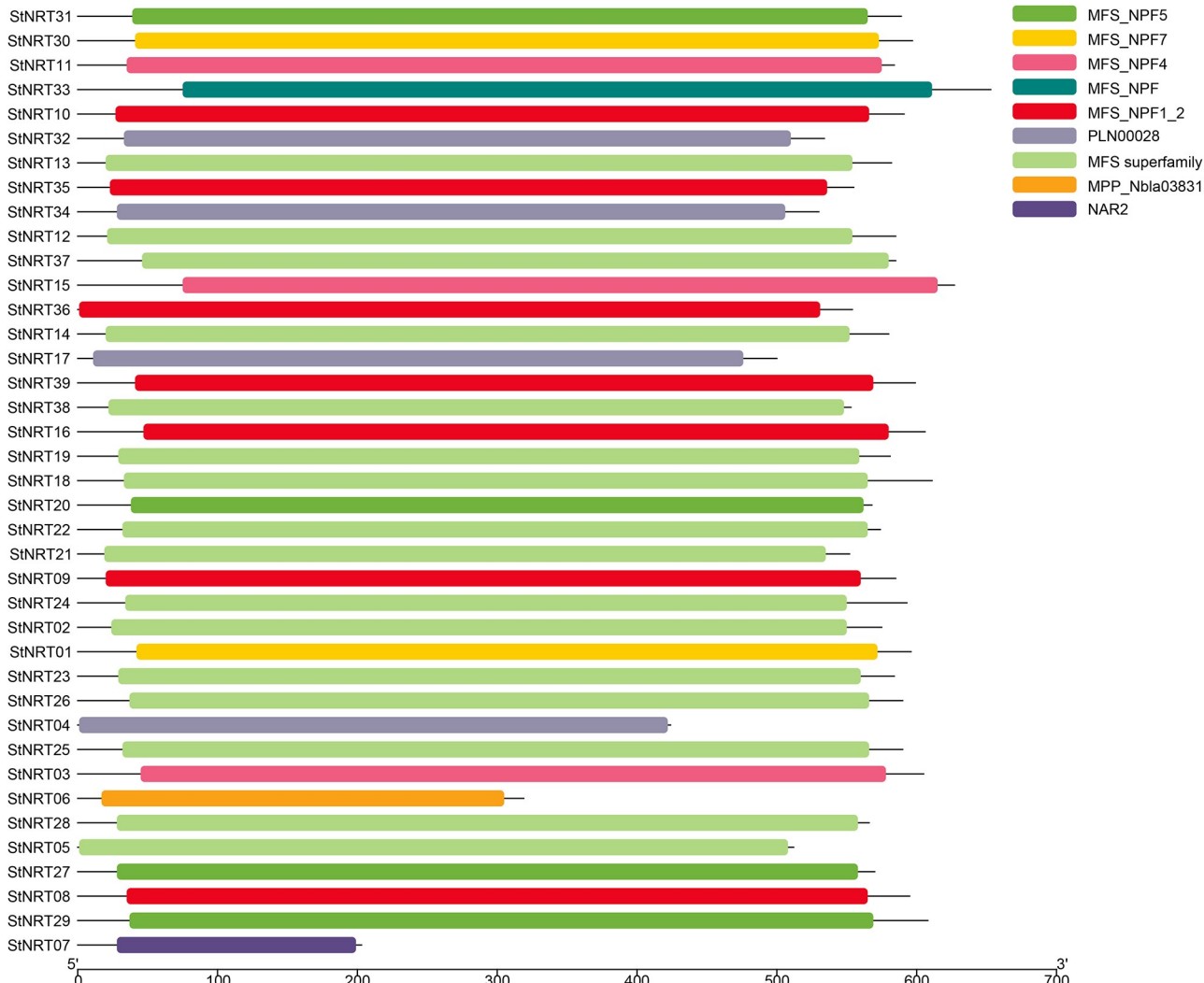

**Fig 1. Conserved domain identification of *StNRT* genes.** The nine type domains are displayed in different colored boxes. The information for each domain is shown on the right. The length of gene/protein can be estimated using the scale at the bottom.

structure, most genes consist of 4 exons (Fig 4b). But some genes are composed of five or more exons, such as *StNRT25*, *StNRT26*, *StNRT03*, *StNRT15*, etc. In addition, there was only one exon found in *StNRT06*.

## Collinearity analysis of *StNRT* members

In order to study the locus relationship between the orthologous of different chromosomes, collinearity analysis was performed. The analysis showed that *StNRT25* and *StNRT26* were highly conserved in chromosome 8. *StNRT08* and *StNRT16* were highly conserved between chromosome 3 and 5 (Fig 5a). For *StNRT* members locus relationship between potato and *Arabidopsis thaliana*, we found that four *StNRT* genes had homologous genes in *Arabidopsis thaliana* (Fig 5b). However, no homologous genes found in *Oryza sativa* (Fig 5c). When comparing potatoes to their near-source species, we found that all *StNRT* members of potato had orthologous genes in eggplant and tomato (Fig 5d and 5e). Especially in tomato, the

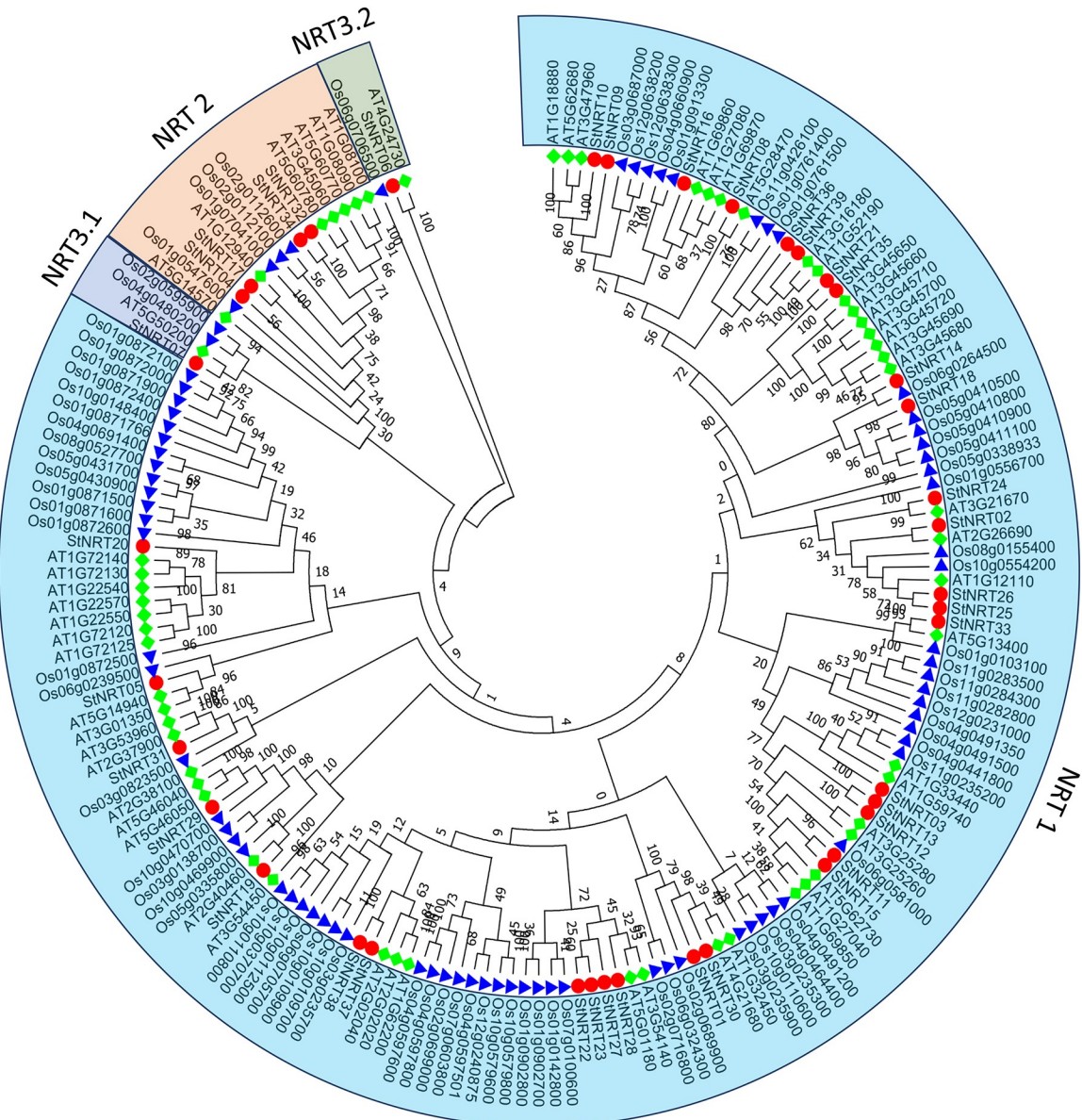

**Fig 2. The phylogenetic tree of potato, rice and *Arabidopsis thaliana* NRT genes.** The 39 potato, 81 rice and 60 *Arabidopsis thaliana* NRT protein sequences were aligned by Clustal X and the phylogenetic tree was constructed using MGA 7.0 by the Maximum likelihood (ML) method. The Bootstrap value was 1,000 replicates. The colored background indicates the different subfamily. Different geometric makers represent NRTs of different plants.

chromosomal position of the orthologous genes of all *StNRT* members was also highly consistent with that of potato.

## Expression patterns of *StNRT* genes in different tissues

Using the RNA-seq data, we showed the expression (FPKM values) of 39 *StNRT* genes in a heatmap in different groups and tissues (Fig 6). The expression results showed that most of the *StNRT* members had a different expression pattern between leaf and root. In addition, the expression of some genes in the nitrogen-deficient group were higher than that in the normal

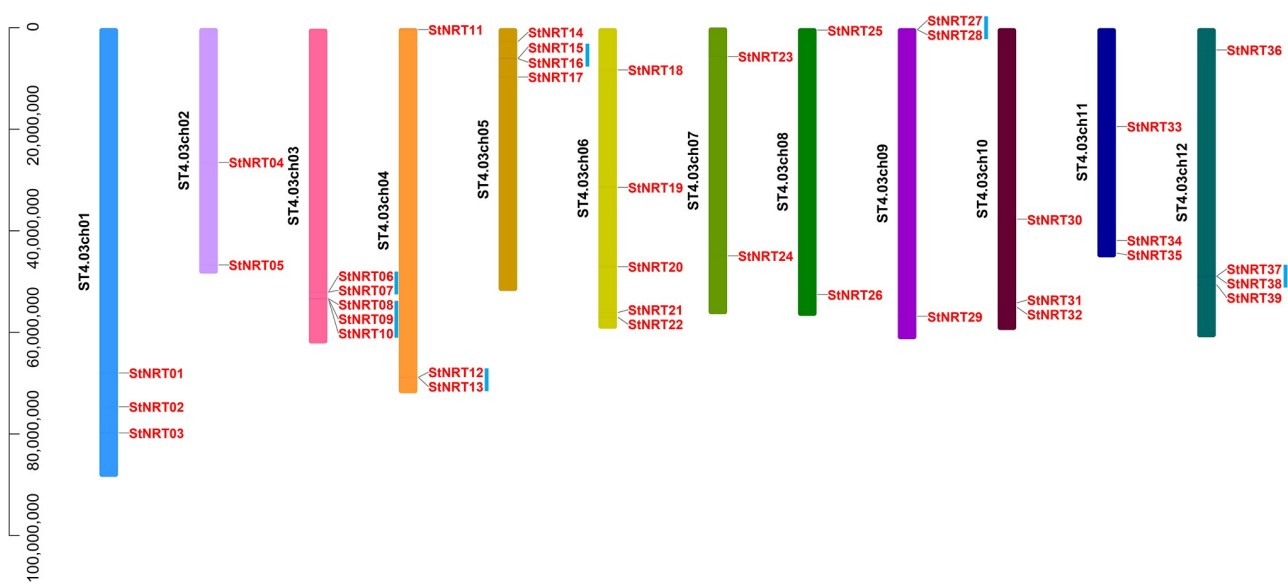

**Fig 3. The chromosomal location and gene duplication of *StNRT* family member in potato.** The ordinate indicates the chromosome length (amino acid length). The tandem duplicated genes are marked by blue rectangles.

nitrogen fertilizer group in root, and the expression profiles of these genes in the leaves are just the opposite. In leaf, *StNRT02* and *StNRT23* were up-regulated in nitrogen-deficient group, but *StNRT27*, *StNRT39*, *StNRT17*, *StNRT26*, *StNRT18*, *StNRT04*, *StNRT05*, *StNRT14*, *StNRT16*, *StNRT24*, and *StNRT37* were down-regulated in nitrogen-deficient conditions. In root, genes like *StNRT35*, *StNRT21*, *StNRT34* were down-regulated by nitrogen-deficient treatment, while *StNRT11*, *StNRT22*, *StNRT30* and *StNRT31* were up-regulated.

### Analysis of Cis-acting element in *StNRT* genes' promoters

After identifying the Cis-acting elements in *StNRT* genes' promoters, we found that MYB, MYC and ERE were the most three elements in all *StNRT* members (Fig 7). *StNRT13* and *StNRT23* had less elements than other members, *StNRT13* contained three elements (MYB, MYC and ERE) and *StNRT23* contained four elements (three MYC and one ABRE). *StNRT*31 (18 elements), *StNRT*26 (17 elements) and *StNRT*18 (16 elements) were the top three genes that contained the most Cis-acting elements.

## Discussion

Nitrate is necessary for plant growth and development. Understanding the gene function and evolution of *NRT* family members is important for plant research. Several studies have elucidated the *NRT* genes functions and evolutionary history in many plant species such as *Arabidopsis thaliana* [17,18], rice [19], poplar [20] and pineapple [21]. In this present study, 39 *StNRT* genes were identified including 33 St*NRT1*, 4 *StNRT2*, and 2 *StNRT3*. Acordding to previous studies, there were 24 *AtNRTs* in *Arabidopsis thaliana* and 48 candidate *NRT* genes in pineapple [18,21]. In total, we identified 39 St*NRTs* in our results, which is within a reasonable range.

As we know, the formation of gene family mainly includes the following ways: 1). whole genome duplication or polyploidization [31]; 2). tandem duplications (of one to a few adjacent genes) [32]; 3). wegmental duplication [33]; 4). transposable elements (TE) [34]; and 5). exon

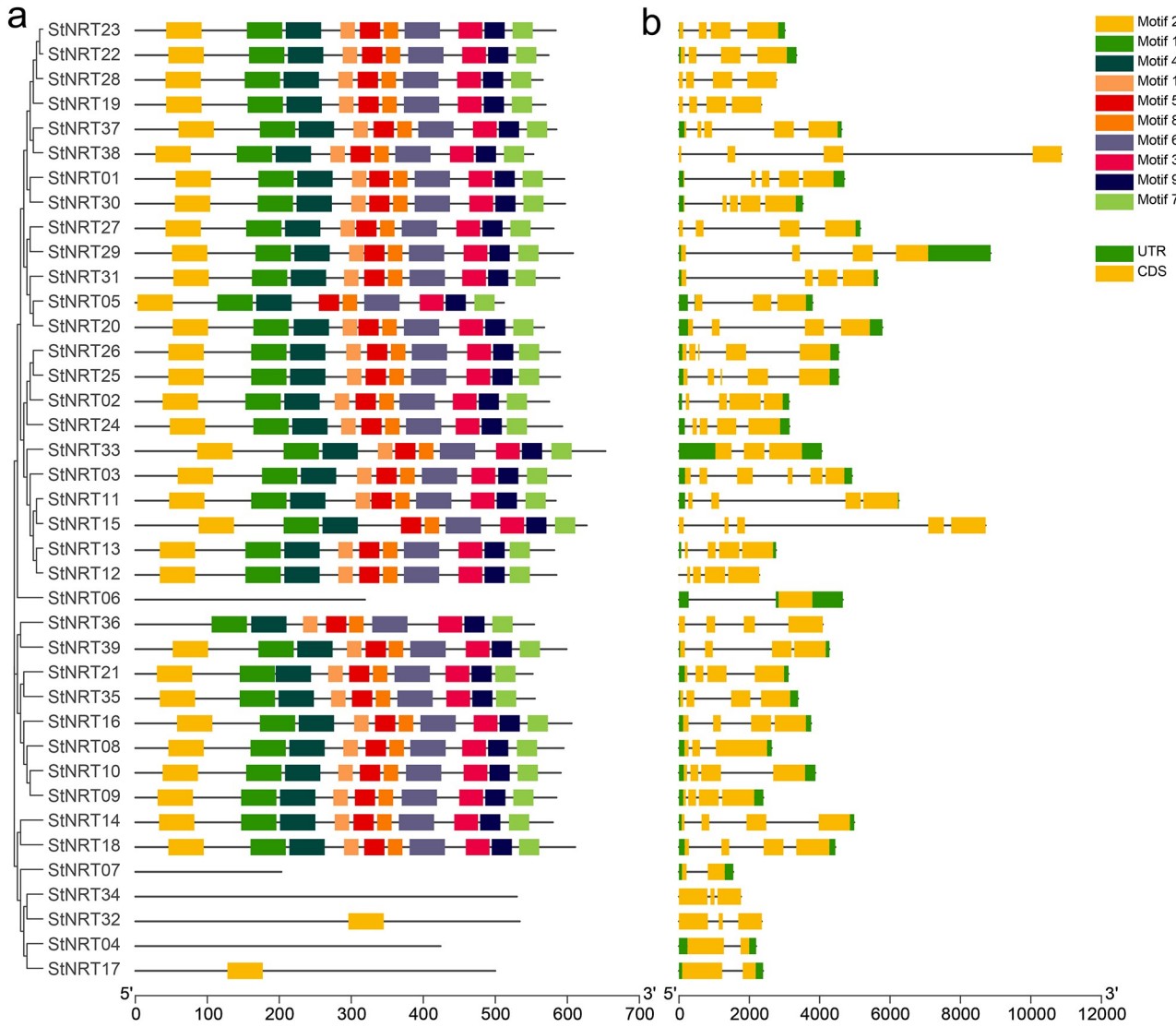

**Fig 4. Conserved motif and gene structure analysis.** (a). Distributions of conserved motifs in *StNRT* members. Ten putative motifs are indicated in different colored boxes. (b). Exon organization of *StNRT* members. Yellow boxes represent exons and black lines with same length represent introns.

duplication and shuffling [35]. In this study, there were 39 *StNRT* members randomly distributed on the 12 potato chromosomes. Of which 13 genes found in 6 duplicated blocks. The gene family members that located in the same block might be formed by tandem duplications. These 13 *StNRT* genes mgiht reveal an early form of gene family member formation. It is speculated that the duplicated genes located in the same block might have closer gene homology, structure and function, which was also confirmed by the evolutionary tree and gene structure analysis in this study. In addition, we found that *StNRT25* and *StNRT26*, *StNRT08* and *StNRT16* are collinear in the potato genome (Fig 5a), indicating that the formation of these genes may be due to segmental duplication or transposable elements.

Gene structure is related its function. Previous studies have shown that there are five conserved domains in the protein sequences of *Arabidopsis thaliana NRT* genes [36], which was consistent with our research. Most of the *NRT* genes are contained in MFS family, which has

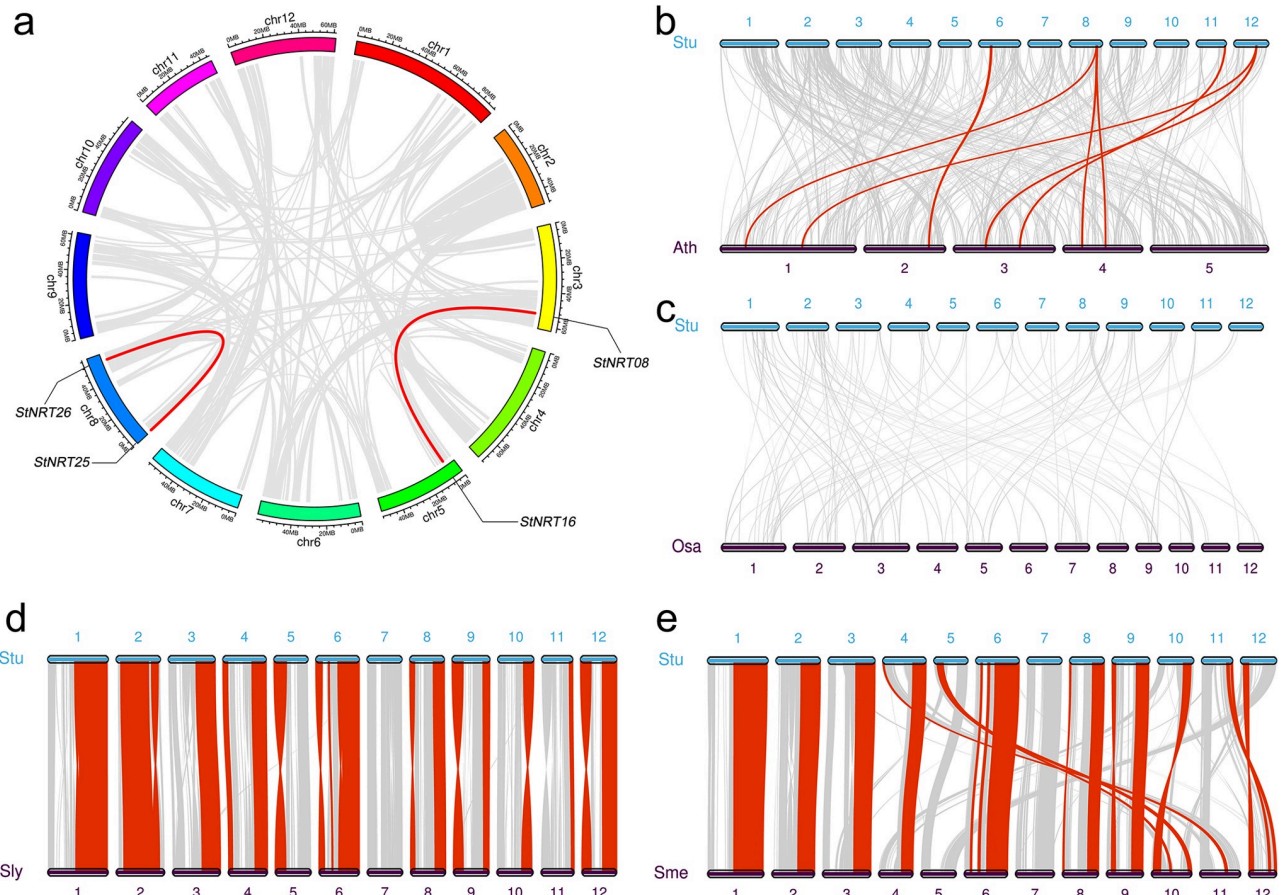

**Fig 5. Collinearity analysis of *StNRT* members.** (a). Collinearity analysis between different chromosomes within the potato genome. Different colors indicate different chromosomes. (b), (c), (d) and (e) showed the collinearity analysis between potato and *Arabidopsis thaliana*, potato and rice, potato and tomato, potato and eggplant, respectively. The red line indicates members of the *StNRT* gene with collinearity, and the gray line indicates other genes. Stu, *Solanum tuberosum*; Ath, *Arabidopsis thaliana*; Osa, *Oryza sativa*; Sly, *Solanum lycopersicum*; Sme, *Solanum melongena*.

12 transmembrane domains [37]. In this study, we found most *StNRT* genes contained MFS family domains. In plants, NRT proteins transport a wide variety of substrates: nitrate, peptides, amino acids, dicarboxylates, glucosinolates, IAA, and ABA [38]. Due to the long intron of *StNRT38* and *StNRT15*, the squence length was greater than other *StNRT* members in potato; moreover, it contained a longer MFS family domain, suggesting that the function of these genes might be more complex. In addition, we found that *StNRT32, StNRT34, StNRT17* and *StNRT04* contained the same domain PLN00028, and these four genes belong to *NRT2* subfamily, indicating that *NRT2* subfamily might works through PLN00028 domian.

The collinearity analysis showed that these *StNRT* members in potato are closely related to *Solanum lycopersicum* and *Solanum melongena*. Especially for *Solanum lycopersicum*, the *NRT* genes also have a good correspondence in the position of the chromosome in both potato and tomato, indicating the close relationship between tomato and potato. These results suggested that *StNRT* family expanded through segmental duplication events during evolution, and the evolutionary events among potato, *Solanum lycopersicum* and *Solanum melongena* might be at an early stage.

Gene expression patterns can provide insights into gene function. Our results showed that most of the *StNRT* members expressed in leaf and root. Some genes were expressed differently

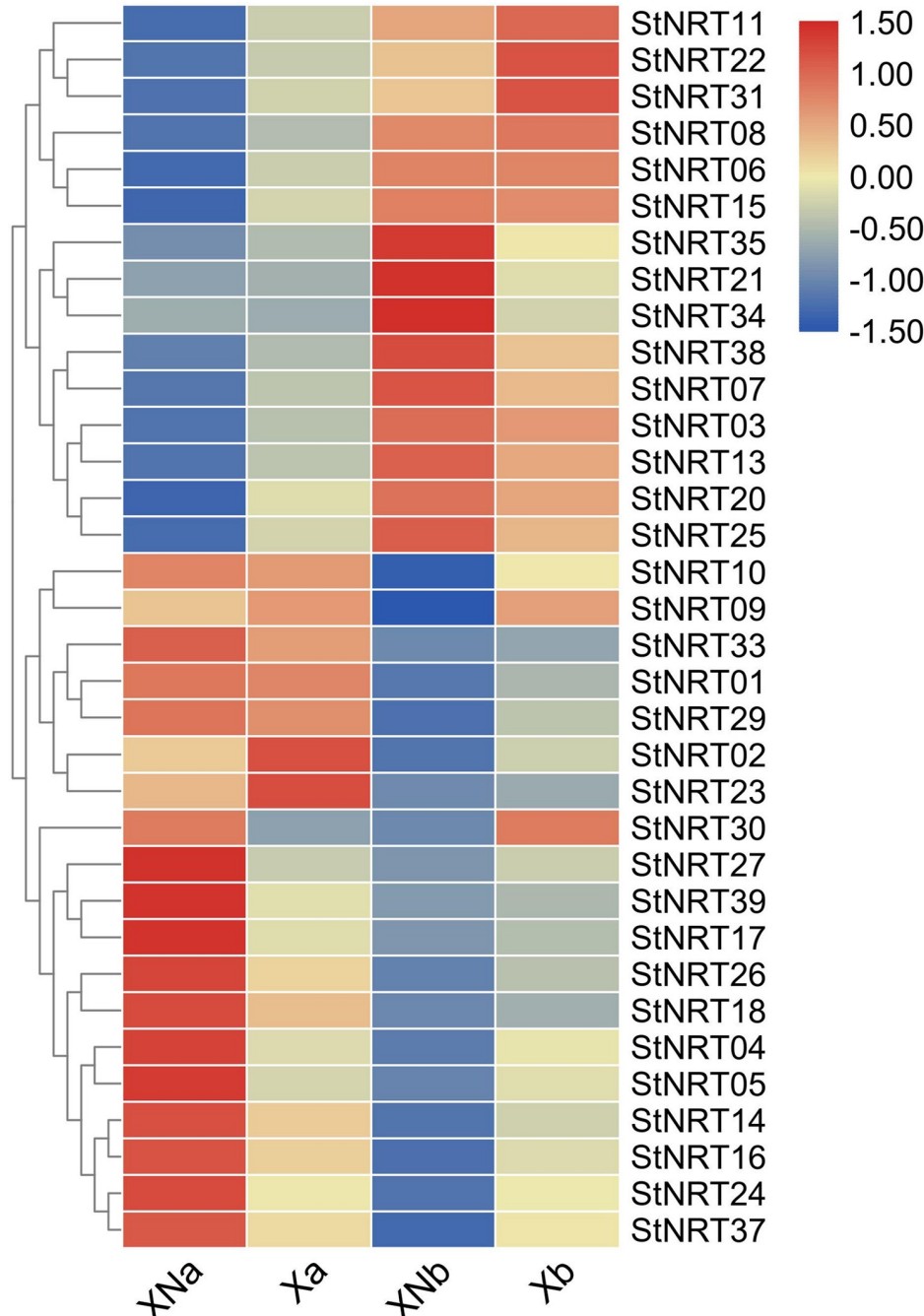

**Fig 6. Expression of *StNRT* members in response to Nitrogen-deficiency.** The Illumina RNA-seq data were downloaded from the SRA database (https://submit.ncbi.nlm.nih.gov/subs/sra/) with the submission number of SRS4186597. Gene expression heatmap obtained by unsupervised comparison of genes differentially expressed in leaf and root. The heatmaps indicate high or low expression levels as green or blue colors, respectively. XN and X represented "Shepody" treated with and without N, respectively. a and b represented leaf and root, respectively.

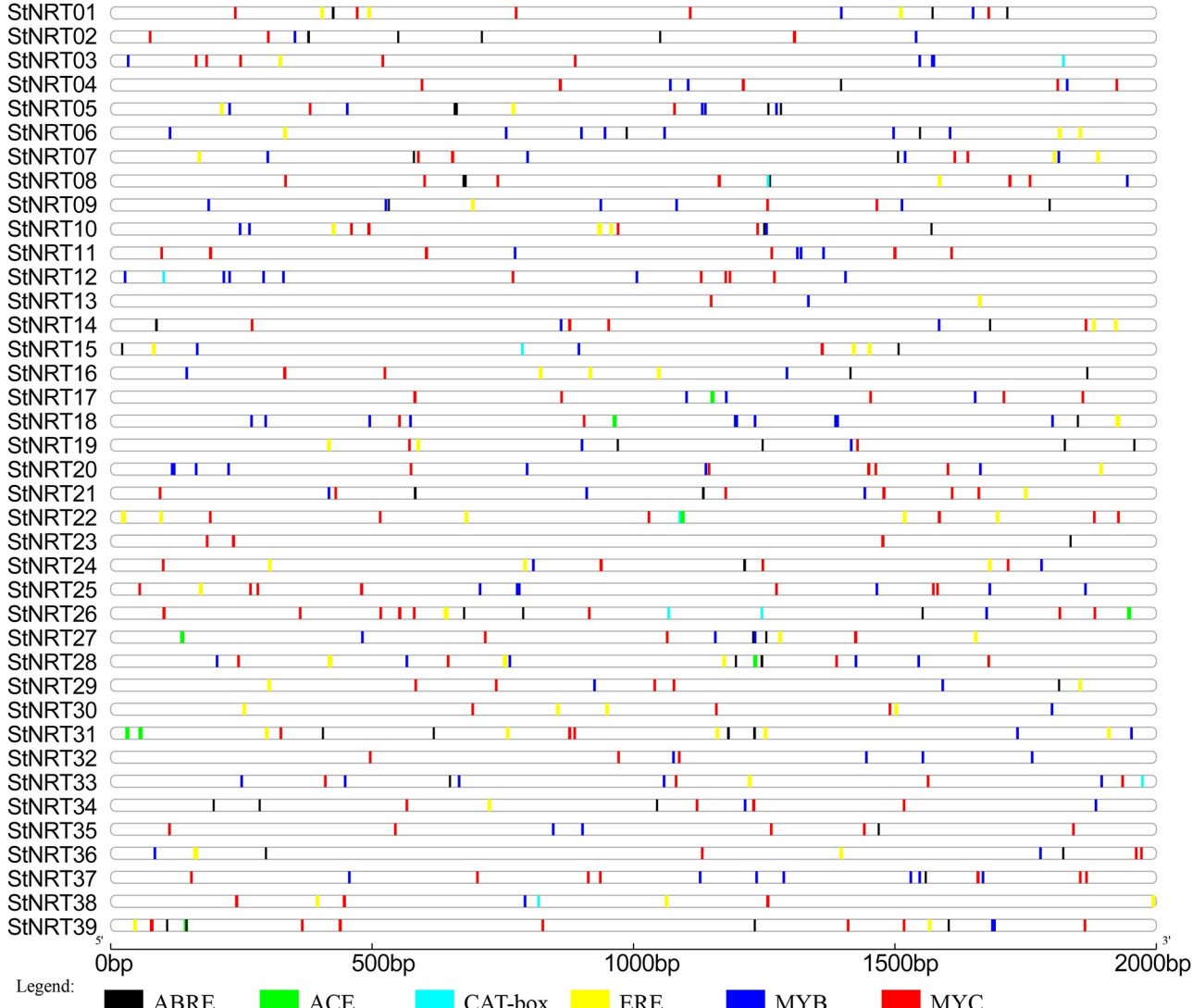

**Fig 7. Predicted cis-elements in *StNRT* promoters.** Promoter sequences (upstream 2000 bp) of 39 *StNRT*20 genes are analyzed by PlantCARE. The upstream length to the translation starts site can be inferred according to the scale at the bottom. Different color boxes represent different elements.

in different organs, such as *StNRT09*, *StNRT10*, *StNRT13*, *StNRT21* etc. Our present study identified that several *StNRT* members were down-regulated by N deficiency (e.g. *StNRT30*, *StNRT17*, *StNRT39*) in leaf, but up-regulated in root. Tiwari et al reported that *StNRTs* were the most down-regulated in roots under low N conditions [39]. According to our previous study, the *NRT* transcripts showed different expression profiles in different potato breeds, especially for varieties with different sensitivity to N deficiency [22]. Hence, we inferred that this might be due to the genetic differences in different potato breeds. However, the different expression profiles indicated that the *NRTs* are crucial for the acquiring N and its conversion to ammonia [40]. *NRT2* family is known to control N uptake and transport and is widely distributed in plants [41]. Lezhneva et al [40] reported that the *Arabidopsis thaliana AtNRT2.5* was only expressed in the shoot and root of *Arabidopsis thaliana* in response to N deficiency. *Arabidopsis thaliana* has 7 *NRT2* family members, and *NRT2.7* is the only *NRT2* member

located on the tonoplast membrane in the seeds, and it functions out of interaction with *NAR2.1* in transporting nitrate [42,43]. However, the expression profiles of *StNRT34*, *StNRT17* and *StNRT04* were decreased in potato leaf by N deficiency, suggesting the increased N metabolism. Different members of *StNRT* play different roles in leaves and roots. Especially under sufficient nitrogen conditions, different members have a clear distribution in different organizations. However, in the Nitrogen-deficiency conditions, all members of the *StNRT* family are widely expressed.

The Cis-acting elements in *StNRT* showed that most of the *StNRTs* might be regulated by TFs like *MYB*, *MYC* and *ERE*. *MYB*, *MYC* and *ERE* are TFs that known to play roles in abiotic stress [44,45]. The widespread recognition site of *MYB*, *MYC* and *ERE* also indicates that these three TFs might be the regulatory factors for *StNRT*. Similarly, Bai et al also found the *MYB* element exists in the promoter region of pepper *NRT* gene [46], which makes our speculation more credible.

## Conclusion

A total of 39 *StNRT* gene family members were identified in the potato genome, including 33 St*NRT1*, 4 *StNRT2*, and 2 *StNRT3*. The collinearity results show that *StNRT* members in potato are closely related to *Solanum lycopersicum* and *Solanum melongena*. For the expression, Different members of *StNRT* play different roles in leaves and roots. Especially under sufficient nitrogen conditions, different members have a clear distribution in different organizations. And most of the *StNRTs* might be regulated by TFs like *MYB*, *MYC* and *ERE*.

## Supporting information

**S1 Fig. Conserved domain identification of *NRT* genes in *Arabidopsis thaliana* and rice.** (a) and (b) shows the Conserved domain identification of *NRT* genes in *Arabidopsis thaliana* and rice, respectively. The abscissa represents the amino acid length. Different colors represent different domains.
(TIF)

**S2 Fig. Conserved motif identification of PLN00028 domain.** Distributions of conserved motifs in *StNRT* members. Ten putative motifs are indicated in different colored boxes.
(TIF)

## Author Contributions

**Conceptualization:** Yu Zhu Han.

**Data curation:** Zhijun Han, Yanfei Zhao, Haoran Ma.

**Formal analysis:** Yanfei Zhao, Yaping Wang.

**Investigation:** Jingying Zhang, Jiayue Zhang.

**Methodology:** Jingying Zhang, Yanfei Zhao, Yaping Wang.

**Software:** Yue Lu, Jiayue Zhang, Haoran Ma.

**Writing – original draft:** Jingying Zhang, Zhijun Han, Yue Lu, Yanfei Zhao, Yu Zhu Han.

**Writing – review & editing:** Jingying Zhang, Zhijun Han, Yue Lu, Yanfei Zhao, Yu Zhu Han.

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
