## [Decision Letter · Decision Letter 0]

29 Dec 2020

PONE-D-20-33972

Genome-wide identification, structural and gene expression analysis of the nitrate transporters (NRTs) family in potato (Solanum tuberosum L.)

PLOS ONE

Dear Dr. Han,

Thank you for submitting your manuscript to PLOS ONE. After careful consideration, we feel that it has merit but does not fully meet PLOS ONE’s publication criteria as it currently stands. Therefore, we invite you to submit a revised version of the manuscript that addresses the points raised during the review process.

We look forward to receiving your revised manuscript.

Kind regards,

Anil Kumar Singh, Ph.D.

Academic Editor

PLOS ONE

Journal Requirements:

Reviewers' comments:

Reviewer's Responses to Questions

**Comments to the Author**

1. Is the manuscript technically sound, and do the data support the conclusions?

Reviewer #1: No

Reviewer #2: Partly

2. Has the statistical analysis been performed appropriately and rigorously? 

Reviewer #1: No

Reviewer #2: N/A

3. Have the authors made all data underlying the findings in their manuscript fully available?

Reviewer #1: No

Reviewer #2: Yes

4. Is the manuscript presented in an intelligible fashion and written in standard English?

Reviewer #1: No

Reviewer #2: No

5. Review Comments to the Author

Reviewer #1: Comments

Line 56-63. The authors are using NRT as well as NTR for referring same gene family. This needs clarification.

Line 65 and Line 288: whatever little is known about potato NRT should be mentioned in introduction, to explain the relevance and novelty of research. The under-mentioned references are a few that are relevant and demand due citation in this MS

Tiwari et al., 2020, Plant Physiol Biochem, 154:171-183

Zhang et al., 2020 PLOS ONE 15(10): e0240662

Pieczynski et al.. 2017, Plant Biotech j 16:603-614

Line 85: The authors should have also included solanacae family also to get a better overview

Line 139-141: The SRA ID given cannot be retrieved from NCBI, if the authors have performed any RNA seq analysis that should be published first to make the data more authentic

Line 165: Consistently the mistake of not italicising Arabidopsis thaliana is done. This is a serious concern it authors warrant publications in prestigious journals like Plos One.

Line 188: Same mistake repeated. It questions the scientific writing skill of the authors.

Line 202: suddenly CIPK gene is mentioned without any prior abbreviations or relevance.

Line 219: Explanation for mentioning PLN00028 domain suddenly is required.

Line 328: very vague justification is supplemented.

Line 342: what is the meaning of TFs, it has been used without prior citation.

Discussion: The section has been prepared with least effort with improper and insufficient references and justification. The importance, relevance, uniqueness of the work is not reflected. Corroboration with previous studies is also not mentioned and contradictory results not justified properly.

Reviewer #2: Authors have identified 39 StNRT gene family members the potato genome. The subfamily classification, gene structure and distribution analysis, and conserved domain prediction for StNRT genes were performed using various bioinformatics tools. Authors downloaded the publicly available RNA seq data and discussed the tissue specific and N2 specific expression of StNRT genes.

Minor comments:

1.Line 24: Nitrate (NO3¯) is the most important source of mineral N for plant growth (rewrite [Nitrogen (N2)]

2. Line 32: For the expression, 33 Different (different) members of StNRT play different roles in leaves and roots.

3. Line 34: different members have a clear distribution in different 35 organizations. (organs?)

4. Line 35: And most of the StNRTs might be regulated by TFs like MYB, MYC and 36 ERE. (rewrite, remove and)

5. Line 45: different organizations. (organs/tissues)

6. Line 35: (organs/tissues)

7. Line 62: Remove reference as text: Dechorgnat et al reported that 62 there were three NTR gene subfamilies: NRT1, NRT2, and NRT3 [16].

8. Line 95: To identified (identify) the domains of the candidate members,

9. Line 102: All the candidate StNRTs were mapped on potato chromosome (chromosomes)

10. Line 104: To identified (identify)

11. Line 105: comparison of all StNRT members were (was) carried

12. rearrange Column 1 in table 1. Make the gene name in one row.

13. Elaborate the Table 1 legend.

14. Elaborate the figure 1 legend.

15. Line 199: Rewrite: Figure 3 showed the location of all these 39 StNRT members, which were 200 randomly distributed on the 12 potato chromosomes.

16. Line 202: Why authors discussed about CIPK or it is just StNRT (clarify), StNRT genes, while chromosome 03 and 06 contains most CIPK genes (5 in each)

17. Line 209: Figure 3 Chromosomal location and gene duplication (mention species, gene family)

18. Line 215: software. The results showed that 10 motifs were identified (Figure 4a). Rewrite

19. Kine 220: we performed motif analysis base on genes of Arabidopsis thaliana (Rewrite)

20. Line 240: Rewrite: we found that all StNRT members in all potato had orthologous genes in

21. Line 227. Figure 4(.) Conserved motif and gene structure analysis

22. Line 245 Figure 5 (.) Collinearity analysis of StNRT members.

23. Line259-261: The expression of StNRT24, 260 StNRT14, StNRT22, StNRT26, StNRT20, StNRT10, StNRT09 and StNRT03 is 261 relatively high. (Which tissue or condition)

24. Line 266: The redder the color, the higher the expression; the bluer the color, the 267 lower the expression. (Write according to previous literature) follow the color range in box.

25. Line 278: Figure 7(.) Predicted cis-elements in StNRT promoters

26. Line 290: Lezhneva reported that there were 24 AtNRTs in Arabidopsis 291 [18] and Li identified 48 candidate NRT genes in pineapple [21] (Remove the named citations and rewrite the sentence)

27. Line 301-302: genes were closely related to each other and had not yet differentiated into fully differentiated gene subfamilies. (use another appropriate word).

Major comments:

1. Check English formatting, grammatical errors.

2. The details and proper condition/treatments of samples used in RNA seq is missing.

3. Verify the tissue specific expression pattern of some differentially expressed StNRT genes, also confirm the N2 specificity of StNRT genes by wet lab experiments.

6. PLOS authors have the option to publish the peer review history of their article (what does this mean?). If published, this will include your full peer review and any attached files.

Reviewer #1: **Yes: **Ragini Sinha

Reviewer #2: **Yes: **Ritesh Kumar

---

## [Author Response · Author response to Decision Letter 0]

23 Jul 2021

Dear Editor and reviewers,

Thank you for your letter and for the reviewers’ comments concerning our manuscript entitled “Genome-wide identification, structural and gene expression analysis of the nitrate transporters (NRTs) family in potato (Solanum tuberosum L.)”. Those comments are all valuable. We have studied comments carefully and made corrections.

The revised manuscript is highlighted in Tracked Changes version.

Reviewer #1: Comments

Line 56-63. The authors are using NRT as well as NTR for referring same gene family. This needs clarification.

Response: We have unified the abbreviation of the full text. Thank you for your comment.

Line 65 and Line 288: whatever little is known about potato NRT should be mentioned in introduction, to explain the relevance and novelty of research. The under-mentioned references are a few that are relevant and demand due citation in this MS

Tiwari et al., 2020, Plant Physiol Biochem, 154:171-183

Zhang et al., 2020 PLOS ONE 15(10): e0240662

Pieczynski et al.. 2017, Plant Biotech j 16:603-614

Response: Thank you for your comment. We have revised this part according to your suggestions. The sentence in the abstract and line-288 are also deleted.

Line 85: The authors should have also included solanacae family also to get a better overview

Response: Thank you for your comment. Your opinions and suggestions are very valuable and worth our attention. However, the aim of this study was to screen the expression levels of NRT gene family members in potato under efficient nitrogen conditions, and to explore the effect of nitrogen deficiency on SRTs. Adding data of other Solanaceae plants can show the evolutionary relationship of NRT gene among different species, which is not consistent with the purpose of this study. But it is indeed a good research direction. Thank you again for your advice.

Line 139-141: The SRA ID given cannot be retrieved from NCBI, if the authors have performed any RNA seq analysis that should be published first to make the data more authentic

Response: Thank you for your comment. Here, we rephrase it to reduce the reader's misunderstanding. This data is uploaded from our previous research

Line 165: Consistently the mistake of not italicising Arabidopsis thaliana is done. This is a serious concern it authors warrant publications in prestigious journals like Plos One.

Response: Thank you for your reminding. We have revised it as required.

Line 188: Same mistake repeated. It questions the scientific writing skill of the authors.

Response: Done as required.

Line 202: suddenly CIPK gene is mentioned without any prior abbreviations or relevance.

Response: Dorry for the basic mistakes. It should be StNRT. 

Line 219: Explanation for mentioning PLN00028 domain suddenly is required.

Response: Thank you for your comment. We have added the explanation for PLN00028.

Line 328: very vague justification is supplemented.

Response: Thank you for your comment. We cited our previous references for discussion. According to our previous study, the NRT transcripts showed different expression profiles in different potato breeds, especially for varieties with different sensitivity to N deficiency [22]. Hence, we inferred that this might be due to the genetic differences in different potato breeds.

Line 342: what is the meaning of TFs, it has been used without prior citation.

Response: Dorry for the basic mistakes. TFs is short for transcription factors

Discussion: The section has been prepared with least effort with improper and insufficient references and justification. The importance, relevance, uniqueness of the work is not reflected. Corroboration with previous studies is also not mentioned and contradictory results not justified properly.

Response: Thank you for your comment. In this round of revision, we have made major changes to the discussion part. We hope this revision will improve the quality of the manuscript.

Reviewer #2: Authors have identified 39 StNRT gene family members the potato genome. The subfamily classification, gene structure and distribution analysis, and conserved domain prediction for StNRT genes were performed using various bioinformatics tools. Authors downloaded the publicly available RNA seq data and discussed the tissue specific and N2 specific expression of StNRT genes.

Minor comments:

1.Line 24: Nitrate (NO3¯) is the most important source of mineral N for plant growth (rewrite [Nitrogen (N2)]

2. Line 32: For the expression, 33 Different (different) members of StNRT play different roles in leaves and roots.

3. Line 34: different members have a clear distribution in different 35 organizations. (organs?)

4. Line 35: And most of the StNRTs might be regulated by TFs like MYB, MYC and 36 ERE. (rewrite, remove and)

5. Line 45: different organizations. (organs/tissues)

6. Line 35: (organs/tissues)

7. Line 62: Remove reference as text: Dechorgnat et al reported that 62 there were three NTR gene subfamilies: NRT1, NRT2, and NRT3 [16].

8. Line 95: To identified (identify) the domains of the candidate members,

9. Line 102: All the candidate StNRTs were mapped on potato chromosome (chromosomes)

10. Line 104: To identified (identify)

11. Line 105: comparison of all StNRT members were (was) carried

12. rearrange Column 1 in table 1. Make the gene name in one row.

13. Elaborate the Table 1 legend.

14. Elaborate the figure 1 legend.

15. Line 199: Rewrite: Figure 3 showed the location of all these 39 StNRT members, which were 200 randomly distributed on the 12 potato chromosomes.

16. Line 202: Why authors discussed about CIPK or it is just StNRT (clarify), StNRT genes, while chromosome 03 and 06 contains most CIPK genes (5 in each)

17. Line 209: Figure 3 Chromosomal location and gene duplication (mention species, gene family)

18. Line 215: software. The results showed that 10 motifs were identified (Figure 4a). Rewrite

19. Kine 220: we performed motif analysis base on genes of Arabidopsis thaliana (Rewrite)

20. Line 240: Rewrite: we found that all StNRT members in all potato had orthologous genes in

21. Line 227. Figure 4(.) Conserved motif and gene structure analysis

22. Line 245 Figure 5 (.) Collinearity analysis of StNRT members.

23. Line259-261: The expression of StNRT24, 260 StNRT14, StNRT22, StNRT26, StNRT20, StNRT10, StNRT09 and StNRT03 is 261 relatively high. (Which tissue or condition)

24. Line 266: The redder the color, the higher the expression; the bluer the color, the 267 lower the expression. (Write according to previous literature) follow the color range in box.

25. Line 278: Figure 7(.) Predicted cis-elements in StNRT promoters

26. Line 290: Lezhneva reported that there were 24 AtNRTs in Arabidopsis 291 [18] and Li identified 48 candidate NRT genes in pineapple [21] (Remove the named citations and rewrite the sentence)

27. Line 301-302: genes were closely related to each other and had not yet differentiated into fully differentiated gene subfamilies. (use another appropriate word).

Response: Thank you very much for your careful review. We are sorry for the basic mistakes we made. We accept your suggestion and have made point-to-point modifications to the manuscript. We hope this revision can improve the quality of the manuscript.

Major comments:

1. Check English formatting, grammatical errors.

Response: Thank you for your comment. We invited our international student Vigo to edit the English, which improved the readability of this manuscript.

2. The details and proper condition/treatments of samples used in RNA seq is missing.

Response: Thank you for your comment. This data is uploaded from our previous research. We cited this article in the place of the data source. In addition, we described the design in the results section and the legend in Figure 6.

3. Verify the tissue specific expression pattern of some differentially expressed StNRT genes, also confirm the N2 specificity of StNRT genes by wet lab experiments.

Response: Thank you for your comment. Due to sample reasons, qRT-PCR experiments cannot be performed. But according to this previously published article [1], qRT-PCR data and RNA-seq data were in excellent consistency, so we believe that this RNA-seq data is reliable.

[1]. Zhang J, Wang Y, Zhao Y, Zhang Y, Zhang J, Ma H, et al. Transcriptome analysis reveals Nitrogen deficiency induced alterations in leaf and root of three cultivars of potato (Solanum tuberosum L.). PLOS ONE. 2020;15(10):e0240662. doi: 10.1371/journal.pone.0240662.

---

## [Decision Letter · Decision Letter 1]

18 Aug 2021

PONE-D-20-33972R1

Genome-wide identification, structural and gene expression analysis of the nitrate transporters (NRTs) family in potato (Solanum tuberosum L.)

PLOS ONE

Dear Dr. Han,

Thank you for submitting your manuscript to PLOS ONE. After careful consideration, we feel that it has merit but does not fully meet PLOS ONE’s publication criteria as it currently stands. Therefore, we invite you to submit a revised version of the manuscript that addresses the points raised during the review process.

We look forward to receiving your revised manuscript.

Kind regards,

Anil Kumar Singh, Ph.D.

Academic Editor

PLOS ONE

Journal Requirements:

Reviewers' comments:

Reviewer's Responses to Questions

**Comments to the Author**

1. If the authors have adequately addressed your comments raised in a previous round of review and you feel that this manuscript is now acceptable for publication, you may indicate that here to bypass the “Comments to the Author” section, enter your conflict of interest statement in the “Confidential to Editor” section, and submit your "Accept" recommendation.

Reviewer #1: (No Response)

Reviewer #2: All comments have been addressed

2. Is the manuscript technically sound, and do the data support the conclusions?

Reviewer #1: Partly

Reviewer #2: Yes

3. Has the statistical analysis been performed appropriately and rigorously? 

Reviewer #1: N/A

Reviewer #2: Yes

4. Have the authors made all data underlying the findings in their manuscript fully available?

Reviewer #1: No

Reviewer #2: Yes

5. Is the manuscript presented in an intelligible fashion and written in standard English?

Reviewer #1: No

Reviewer #2: Yes

6. Review Comments to the Author

Reviewer #1: Prior to the acceptance, the authors needs to rectify the MS for grammatical and scientific errors, like use of botanical names at all places and missing italics at some places. In the abstract the authors have mentioned, evolutionary relationship as a scope of the study, however, in the justification to reviewers, they have mentioned that evolutionary studies is beyond the scope of the study. This needs to be rectified.

lFewexamples:i

9line no. 29-30: proper meaning is not reflected.

Table 1: use pI; PI is not the proper notion.

line 204: MEGA 7.0, incorrectly written

Reviewer #2: Authors have carefully edited the manuscript and addressed all previous concerns. Authors should proof read the manuscript one more time before publishing.

7. PLOS authors have the option to publish the peer review history of their article (what does this mean?). If published, this will include your full peer review and any attached files.

Reviewer #1: No

Reviewer #2: **Yes: **Ritesh Kumar

---

## [Author Response · Author response to Decision Letter 1]

27 Aug 2021

Dear Editor and reviewers,

Thank you for your letter and for the reviewers’ comments concerning our manuscript entitled “Genome-wide identification, structural and gene expression analysis of the nitrate transporters (NRTs) family in potato (Solanum tuberosum L.)”. Those comments are all valuable. We have studied comments carefully and made corrections.

The revised manuscript is highlighted in Tracked Changes version.

Reviewer #1: Comments

Comments:

Prior to the acceptance, the authors needs to rectify the MS for grammatical and scientific errors, like use of botanical names at all places and missing italics at some places. In the abstract the authors have mentioned, evolutionary relationship as a scope of the study, however, in the justification to reviewers, they have mentioned that evolutionary studies is beyond the scope of the study. This needs to be rectified.

Response: thank you for your comments. First of all, thank you for your recognition. We further revised the graphical and scientific errors as required. Also, we revised the scope in the abstract section.

Comments:

line no. 29-30: proper meaning is not reflected.

Response: thank you for your comments. We rephased the sentence to “Totally, 39 StNRT gene members were identified in potato genome, including 33, 4 and 2 member belong to NRT1, NRT2, and NRT3, respectively.”

Comments:

Table 1: use pI; PI is not the proper notion.

line 204: MEGA 7.0, incorrectly written

Response: thank you for your comments. Both the two errors were revised as required.

Reviewer #2: Comments

Comments:

Authors have carefully edited the manuscript and addressed all previous concerns. Authors should proof read the manuscript one more time before publishing.

Response: Thank you for your recognition. We have revised the grammatical and scientific errors of the manuscript. Thank you again for your review.

---

## [Editor Report · Decision Letter 2]

1 Sep 2021

Genome-wide identification, structural and gene expression analysis of the nitrate transporters (NRTs) family in potato (Solanum tuberosum L.)

PONE-D-20-33972R2

Dear Dr. Han,

We’re pleased to inform you that your manuscript has been judged scientifically suitable for publication and will be formally accepted for publication once it meets all outstanding technical requirements.

Kind regards,

Anil Kumar Singh, Ph.D.

Academic Editor

PLOS ONE
---

## [Editor Report · Acceptance letter]

13 Oct 2021

PONE-D-20-33972R2 

Genome-wide identification, structural and gene expression analysis of the nitrate transporters (*NRTs*) family in potato (Solanum tuberosum L.) 

Dear Dr. Han:

I'm pleased to inform you that your manuscript has been deemed suitable for publication in PLOS ONE. Congratulations! Your manuscript is now with our production department. 

Kind regards, 

on behalf of

Dr. Anil Kumar Singh 

Academic Editor

PLOS ONE